# SYNTHETIC CONTROLS AS BALANCING SCORES

**Harsh Parikh**
Duke University
harsh.parikh@duke.edu

## ABSTRACT

We outline the factors under which conditioning on Synthetic Control (SC) weights emulates a randomized control trial where the treatment status is independent of potential outcomes. Specifically, we demonstrate that if there exist SC weights such that the treatment effects are exactly identified, and these weights are uniformly and cumulatively bounded, then *SC weights are balancing scores*.

## 1 INTRODUCTION

Synthetic controls are one of the popular methods in causal inference on time-series data where the population (or sample size) is fairly smaller compared to the dimensionality of the data Ferman & Pinto (2017). For instance, synthetic control methods were used to study the effect of the reunification of Germany on West Germany's GDP per capita Abadie et al. (2015); Shi et al. (2021). A synthetic West Germany under no-reunification was constructed using the linear combination of the GDP per capita data from 16 other OECD countries Abadie et al. (2015). Synthetic control is widely used in the social science literature because of the simplicity of implementation and analysis Ferman & Pinto (2016; 2017). Here, one typically regresses the pre-treatment outcomes of non-treated units on the pre-treatment outcomes of a treated unit to find the set of linear weights. These linear weights are used to estimate the counterfactual outcome Abadie et al. (2010).

On the other hand, balancing score methods such as propensity and prognostic scores have been widely used to adjust for confounding in situations where the number of units in the sample is larger than the number of features Rosenbaum & Rubin (1983); Hansen (2008). For instance, propensity score adjustments were used to study the treatment effect of coronary artery bypass surgery Rosenbaum & Rubin (1983); Cohn et al. (1981). These methods are known as balancing score methods as it is sufficient to condition one of these scores to emulate a randomized trial Austin (2011).

Synthetic control and balancing score methods are applicable in fairly distinct scenarios. However, they are similar in terms of their simplicity, ease of use, and the use of pretreatment covariates to learn parameters that aid in treatment effect identification. The theoretical properties of these approaches are well-studied in the literature (Abadie et al., 2015; Ferman & Pinto, 2016; 2017; Shi et al., 2021; Rosenbaum & Rubin, 1983; Hansen, 2008; Parikh et al., 2022). However, the connections between them are not well understood.

This short paper bridges this gap by showing that SC weights are balancing scores under a sufficient set of assumptions (Section 3). Our result is of general interest because conditioning on balancing scores guarantees ignorability (i.e., that potential outcomes and treatment assignment are mutually independent) – ensuring consistent estimation of treatment effects (Johnson, 2013). The result presented in this paper enables sharing of theoretical and methodological advancements across these two broad approaches - hence, advancing the literature and its applicability to complex real-world problems.

## 2 PRELIMINARIES

Consider a finite population of $n$ units, such that for each unit $i$, we observe: (i) $Z_i$, the binary treatment indicator, (ii) $\{X_{i,t}\}_{t=0}^{t_0}$, the time-series of pre-treatment outcomes, and (iii) $\{Y_{i,t}\}_{t=t_0+1}^{T}$, the time-series of post-treatment outcomes. Similar to SC literature, we consider the case with exactly one treated unit. Without loss of generality, let $Z_1 = 1$ and $\forall j > 1, Z_j = 0$. We assume that these

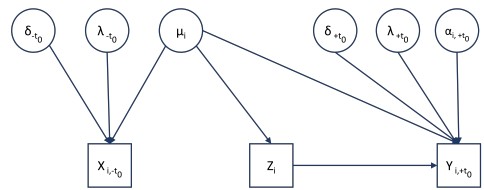

Figure 1: A simplified causal directed acyclic graph for the discussed exposition. Here, with the slight abuse of notation, $-t_0$ refers to the time before intervention and $+t_0$ refers to the time after the intervention.

time-series are generated using the following factor model:

$$\text{for } t \leq t_0 : \qquad X_{i,t} = \delta_t + \lambda_t \mu_i + \epsilon_{i,t} \tag{1}$$
$$\text{for } t > t_0 : \qquad Y_{i,t} = \delta_t + \lambda_t \mu_i + Z_i \alpha_{i,t} + \epsilon_{i,t} \tag{2}$$

where, (i) $\delta_t$ is an unobserved common time-trend across units, (ii) $\mu_i$ is an unobserved unit-specific factor, (iii) $\lambda_t$ is the time-specific factor loading, (iv) $\epsilon_{i,t} \sim \mathcal{N}(0, \sigma_{i,t}^2)$ is the noise at time $t$ and (v) $\alpha_{i,t}$ is the treatment effect for unit $i$ at time $t$. We are interested in identifying $\{\alpha_{1,t}\}_{t=t_0+1}^T$. This setup is similar to the one discussed in Ferman & Pinto (2017).

We will use bold letters to denote the collection of observations across units: $\boldsymbol{Z} = \{Z_i\}_{i=1}^n$, $\boldsymbol{X} = \{\{X_{i,t}\}_{t=0}^{t_0}\}_{i=1}^n$ and $\boldsymbol{Y} = \{\{Y_{i,t}\}_{t=t_0+1}^T\}_{i=1}^n$. Consider, the scenario where the probability of observing treatment assignments is a function of the unit-specific factors $\boldsymbol{\mu} = \{\mu_i\}_{i=1}^n$ i.e. $\boldsymbol{Z} \sim f(\boldsymbol{\mu})$ for some (unknown) distribution $f$ such that $\boldsymbol{1}^T \boldsymbol{Z} = 1$. Figure 1 graphically demonstrates the structural causal dependencies.

Consider the following assumptions key for identification of causal effects (discussed in Appendix A):
*A.1. (**feasibility**)* $\exists \boldsymbol{\beta}$ s.t. $\mathbb{E}\left(X_{1,t} - \sum_{i=2}^n \beta_i X_{i,t}\right) = 0$.
*A.2. (**uniformly bounded**)* $\forall i$, $0 \leq \beta_i \leq 1$,
*A.3. (**cumulatively bounded**)* $\sum_{i=2}^n \beta_i = 1$ *i.e.* $\boldsymbol{1}^T \boldsymbol{\beta} = 1$.

## 3 RESULT: SC WEIGHTS ARE BALANCING SCORE

This section discusses the concept of balancing score and our main result. For our setup, if there exists a function $b$ such that $\{\boldsymbol{Y}_t(\boldsymbol{z})\}_t \perp \boldsymbol{Z} | b(\mathbf{X})$ then $b$ is a *balancing score*. However, consider the causal graph in Figure 1: it is not obvious that $b(\mathbf{X}) = \mathbf{X}$ is a balancing score.

We know that $\{\boldsymbol{Y}_t(\boldsymbol{z})\}_t \perp \boldsymbol{Z} | \boldsymbol{\mu}$; thus, $\boldsymbol{\mu}$ is a balancing score. However, one must note that $\boldsymbol{\mu}$ is unobserved and hence it is not a very useful balancing score. In this discussion, we show that the (oracle) SC weights are also balancing scores i.e. $\{\boldsymbol{Y}_t(\boldsymbol{z})\}_t \perp \boldsymbol{Z} | \boldsymbol{\beta}$.

**Theorem 1** *Given the causal dependencies and assumptions A.1 - A.3, $\{\boldsymbol{Y}_t(\boldsymbol{z})\}_t \perp \boldsymbol{Z} \mid \boldsymbol{\beta}$.*

We provide the proof of this theorem in Appendix B.

## 4 CONCLUSION AND DISCUSSION

In this short paper, we discuss the set of sufficient conditions under which SC weights are balancing scores i.e. the potential outcomes are independent of treatment assignment. Balancing score is an important property that allows methods like propensity and prognostic to emulate randomized control trials. Our result implies that conditioning on SC weights helps emulate a controlled trial where one of the $n$ units is treated.

This understanding allows for a deeper understanding of the validity of SC inferences and furthering the methodological development. Doubly robust methods that combine propensity and prognostic scores are common in cross-sectional data. Our future work wil focus on developing a doubly robust SC method that allows for efficient and robust inference. Further, more work is needed to understand if this result holds in different scenarios (under weaker assumptions) where synthetic controls have been used for estimation.

## URM Statement and Acknowledgments

The author thanks the ICLR Tiny Papers Track for this unique opportunity. Harsh Parikh meets the URM criteria of the ICLR 2023 Tiny Papers Track. The author wants to thank Amazon Science Fellowship and NSF grant IIS-2147061 for supporting their research.

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

## A   Identification of Treatment Effect using Synthetic Controls

Assumption A.1. implies that $\mu_1 = \sum_{i=2}^{n} \beta_i \mu_i$. If we choose $\beta_i$'s such that the condition in A.1. are satisfied, then

$$\mathbb{E}\left(Y_{1,t} - \sum_{i=2}^{n} \beta_i Y_{i,t}\right) = \lambda(\mu_1 - \sum_{i=2}^{n} \beta_i \mu_i) + \alpha_{1,t} = \alpha_{1,t}.$$

Thus, identifying $\boldsymbol{\beta}$ such that $\mathbb{E}\left(X_{1,t} - \sum_{i=2}^{n} \beta_i X_{i,t}\right) = 0$ leads to the identification of the treatment effect $\alpha_{1,t}$.

Note that, we refer to $\beta$'s as *oracle weights* because it may not be feasible to exactly identify them using the observed finite population, especially, when the noise is heteroskedastic[1] (Ferman & Pinto, 2017). For the rest of the argument, we will assume that we have knowledge of these oracle weights. However, one can always estimate approximate oracle weights by fitting a regularized linear regression on the pre-treatment outcomes (Abadie et al., 2010; 2015).

---

[1]To estimate, these weights, in practice, one fits a model that uses the outcomes of the control units to predict the contemporary outcome of the treated unit in the pretreatment period: `regress` $\mathtt{y}=X_{i,t}$, $\mathtt{x}=\{X_{2,t} \ldots X_{n,t}\}$. Typically, the regularization ensures that the weights are bounded between 0 and 1 and the sum of the weights is one.

## B  PROOF OF THEOREM 1

**Proof.** Consider an $n \times n$ matrix $\mathbf{B}$ such that: (a) $\mathbf{B}_{i,i} = 0$, (b) $\mathbf{B}_{i,1} = 1$ (for $i > 1$), (c) $\mathbf{B}_{1,j} = \beta_j$ (for $j > 1$), and (d) $\mathbf{B}_{i,j} = -\beta_j$ (for $j > 1$ and $i > 1$). Now, *we will show*, $\{Y_t(z)\}_t \perp Z | \mathbf{B}$. We know that $\mathbb{E}(\boldsymbol{X}_t) = \delta_t + \lambda_t \boldsymbol{\mu}$, $\mathbb{E}(X_{1,t}) = \sum_{i=2}^{n} \beta_i \mathbb{E}(X_{i,t})$ and $\sum_{i=2}^{n} \beta_i = 1$. These results imply that $\mu_1 = \sum_{i=2}^{n} \beta_i \mu_i$. We now observe that $\boldsymbol{\mu} = \mathbf{B}\boldsymbol{\mu}$, i.e. $\boldsymbol{\mu}$ is one of the eigenvectors of the matrix $\mathbf{B}$ with corresponding eigenvalue equal to 1. Hence, $\{Y_t(z)\}_t \perp Z | \mathbf{B}$ and $\mathbf{B} = g(\boldsymbol{\beta})$ where $g$ is the discussed matrix construction. *QED.*

*Note* that as $\boldsymbol{\beta}$ is a deterministic function of $\mathbf{X}$, $\mathbf{X}$ is also a balancing score i.e. $\{Y_t(z)\}_t \perp Z | \mathbf{X}$.

