# OpenReview forum: "Synthetic Controls as Balancing Scores"
_ICLR.cc/2023/TinyPapers — Submitted to Tiny Papers @ ICLR 2023_

### Official Review · Reviewer_otMo · 2023-03-30

**Confidence:** 2

**Summary Of Contributions:**

Synthetic control and balancing score methods are techniques used for estimating the causal effect of a treatment. In this work, the author demonstrates that, under certain assumptions, synthetic control weights can be interpreted as balancing scores.

**Rating:**

Clear, Correct, and Reproducible (CCR): a submission which meets the reviewing criteria

**Strengths And Weaknesses:**

- **Strengths**
    - The paper is well-written and well-structured, with appropriate references. The section on synthetic control, including the setup, assumptions, and identification, is particularly strong and provides the reader with a clear understanding of the mathematical foundations. The mathematical background is supported by helpful visuals, such as Figure 1a and 1b, which make the content easier to digest for the reader.
    - The author's research presents a compelling connection between two methods by showing how synthetic control weights can be viewed as balancing scores
    - The assumption of having knowledge of the oracle weights serves to reinforce the Synthetic Control assumptions and identification.
- **Weakness**
    - I would encourage the author to provide more details on the scenarios in which synthetic control weights cannot be interpreted as balancing scores, along with the relevant assumptions and a mathematical foundation for these cases. Such an expansion would enable other researchers to build on this work.

**Suggested Changes:**

I suggest that the "Synthetic Control: Setup, Assumptions and Identification" section be structured under a new section titled "Methods" section, for better readability.

---

### Official Review · Reviewer_fquT · 2023-04-04

**Confidence:** 2

**Summary Of Contributions:**

The authors in this paper proposes that under specific conditions, the synthetic control weights csn be treated as balancing scores.

**Rating:**

Clear, Correct, and Reproducible (CCR): a submission which meets the reviewing criteria

**Strengths And Weaknesses:**

Strengths
* The paper is well-written and well-structured. The author perfectly sets up the mathematical formulation of the problem and uses figures to help the reader to digest the content.

Weakness
* The author does not discuss the practical implications of the assumptions and the conclusion of this paper.
* The section titles are not numbered.

**Suggested Changes:**

* The author should discuss the practical implications of the assumptions and the conclusion of this paper.
* The section titles should be numbered.

---

### Comment · Area_Chair_TY7S · 2023-05-30
**Invite to archive**

Hi All,

This paper was invited to archive.  The reviews and meta reviews were broadly positive, and so I believe this paper should still be archived.

As a general comment:  I do not believe the author(s) updated the manuscript in light of reviewer feedback.  I would like to emphasise to the author(s) that incorporating pertinent reviewer feedback is an important part of the publication process, and nearly always leads to a better paper (more easily understood, more usable, more impact, more citations etc).

Congratulations, and good work.
AC TY7S

---

> ### Author Response · Authors · 2023-05-31
> **Thank you so much for the feedback**
>
> Dear Area Chair, Program Chair, and Reviewers,
> We are extremely thankful to you for your thoughtful and constructive comments. We have made changes as per your suggestions to make the paper less dense and more accessible. Particularly, we have added two paragraphs introducing synthetic controls and balancing scores in the introductions. We have moved the theoretical proofs to the appendix while keeping the main results in the main text. We have also added a short discussion on the implications of our results and future work.
> We will love to opt-in to archive our work with ICLR. Thank you so much, again!

---

> > ### Comment · Area_Chair_TY7S · 2023-06-02
> > **URM Statement**
> >
> > Thank you for updating the manuscript.  Please remember to include the URM statement as well prior to archival.  See the advice in the template:
> >
> > ```Please include this URM Statement section at the end of the paper but before the references before. In your anonymized submission, we recommend stating ``The authors acknowledge that at least one key author of this work meets the URM criteria of ICLR 2023 Tiny Papers Track.'' For the camera ready version, we ask authors to identify which author(s) meet the URM criteria, e.g., ``Author TFB meets the URM criteria of ICLR 2023 Tiny Papers Track.'' The authors are also welcome to come up with their own phrases to affirm meeting this criteria.```

---

> > > ### Author Response · Authors · 2023-06-02
> > > **Unable to update the submission with URM statement**
> > >
> > > I added the URM statement but I am unable to update the submission

---

> > > > ### Comment · Area_Chair_TY7S · 2023-06-03
> > > > **Re-opened**
> > > >
> > > > The submissions have been temporarily re-opened for you to re-upload a revised version.  Thank you very much.

---

> > > > > ### Author Response · Authors · 2023-06-03
> > > > > **Thanks! I uploaded the new version with URM statement**
> > > > >
> > > > > Thanks! I uploaded the new version with URM statement

---

### Comment · Area_Chair_TY7S · 2023-06-06
**Ready for archival**

This work meets the threshold for archival, contents the URM statement and is deanonymized.

---

### Meta-Review · Area_Chair_TY7S · 2023-04-06

**Recommendation:** Invite to present
**Confidence:** 3

**Metareview:**

This paper tries to connect two methods in casual inference.  The paper itself is well written, especially in the style of a short technical note.  It is very dense, without much assistance for readers that aren't experts in the field (myself included...).  This will put some readers off (but maybe there isn't much for those readers in this level of depth anyway, and the core result is sufficient).  The work appears novel, and could be significant to the field.

**Summary:**

Under some reasonable assumptions, two core methods in CI are related.  The paper is very dense.

**Comments And Feedback To The Authors:**

Please take on board and implement the feedback from the reviewers.  Please also make use of an appendix to defer some of the technical content to make space for even a few sentences on the high-level significance, opportunities, implications, use-cases etc.  This will make the paper much more impactful and wide-reaching.  Otherwise, good work.

**Reason For Not Giving A Higher Recommendation:**

Unfortunately I do not think I am sufficiently experienced in this field to be confident that this work should be deemed as "notable", even though it is definitely good enough to be included.  If some content were trimmed to a supplement, and some extra space dedicated to make the paper more accessible, then I think this could be a very strong submission.

**Reason For Not Giving A Lower Recommendation:**

I think both reviews gave quite low scores for the (low) level of criticism they had.  Their scores may have been tempered by their low indicated confidence.

---

### Decision · Program_Chairs · 2023-04-10

Invite to archive